# Carotid Web as a Cause of Ischemic Stroke: Effective Treatment with Endovascular Techniques

**DOI:** 10.3390/jcm14082568

**Published:** 2025-04-09

**Authors:** Magdalena Konieczna-Brazis, Pawel Brazis, Milena Switonska, Arkadiusz Migdalski

**Affiliations:** 1Department of Neurology and Clinical Neurophysiology, Jan Biziel University Hospital No. 2, Collegium Medicum in Bydgoszcz, Nicolaus Copernicus University in Torun, Poland, Ujejskiego Street 75, 85-168 Bydgoszcz, Poland; magdabrazis@gmail.com (M.K.-B.); m.switonska@cm.umk.pl (M.S.); 2Department of Vascular Surgery and Angiology, University Hospital No. 1, Collegium Medicum in Bydgoszcz, Nicolaus Copernicus University in Torun, Poland, M. Sklodowskiej-Curie Street 9, 85-094 Bydgoszcz, Poland; armigos@wp.pl

**Keywords:** carotid web, ischemic stroke, thrombectomy, carotid artery stenting, endovascular

## Abstract

**Background**: Carotid web (CaW) usually presents as a shelf-like intimal flap at the beginning of the internal carotid artery. It has been proven that CaW is associated with ischemic stroke, particularly in young patients without other risk factors. This case report aimed to describe the carotid web that causes ischemic stroke due to embolic complications. Moreover, both pathologies were successfully treated with endovascular techniques in the presented case study. **Methods**: A 59-year-old male presented to the neurological department with motor aphasia, right-sided weakness, and hypoesthesia. Computer tomography (CT) of the head and computed tomography angiography (CTA) of the aortic arch and intracranial arteries were performed. Due to the unknown onset of the presented stroke symptoms, diagnostics were extended to magnetic resonance (MR), and based on this, the patient qualified for immediate mechanical thrombectomy (according to the DAWN trial protocol). Intraoperative digital subtraction angiography (DSA) revealed embolism material in the left middle cerebral artery (segment M1). The artery was recanalized via aspiration thrombectomy using the Penumbra system, and complete restoration of flow was obtained (according to the TICI scale). In addition, DSA revealed the presence of CaW changes in the left internal carotid artery (LICA). In the control CT scanning, an acute ischemic area in the left temporal lobe was found. After the treatment, the patient demonstrated complete neurological improvement from his initial presentation. He qualified for carotid artery stenting of the LICA, which was postponed to a later period due to the presence of an area of infarction. The angioplasty with stenting was performed 6 months later, and a carotid antiembolic “mesh” stent (Roadsaver, Terumo) was implanted into the LICA across the carotid web. **Conclusions**: CaW should be considered in the case of stroke resulting from unknown causes. The presented case study demonstrated that both carotid web and ischemic stroke pathologies can be effectively treated with emerging endovascular techniques.

## 1. Introduction

Carotid web (CaW), also known as carotid diaphragm, is an unusual variant of fibro-muscular dysplasia (FMD). CaW is specified by the fibrous intimal flap, located on the posterior wall of the carotid bulb. The development of thrombogenesis behind the intimal flap is usually associated with an increased risk of cerebral embolic stroke [1].

Although the incidence of CaW is rare, a strong relationship between this change and stroke has been shown, especially in young patients without other vascular conditions. A study showed that CaW occurred in 7/576 (1.2%) patients with suspected stroke during a hospital-based period and indicated that it may be a reasonable expectation in 2.5–37% of cryptogenic strokes [2,3,4,5,6]. CaW is also characterized by a high percentage of recurrent stroke, with a rate of about 27% at 5 years if conservative treatment is used [5].

Endovascular treatment significantly reduces the risk of recurrent stroke. In such cases, angioplasty with stenting is a notably minimally invasive and relatively safe procedure. There is no significant narrowing of the vessel in the place of the CaW, which significantly facilitates the passage of the guidewire and stent implantation [7].

We present a CaW case complicated by acute ischemic stroke of the left-brain hemisphere as a result of the occluded middle cerebral artery (MCA). In the first stage, the patient was treated with intracranial thrombectomy, and then in the second stage, carotid angioplasty with stenting (CAS) was performed. The treatment strategies applied in both phases were shown to be effective and promising.

## 2. Case History

A 59-year-old male with a history of arterial hypertension and hyperlipidemia was transferred to our Department of Neurology at the Center of Interventional Treatment of Stroke due to motor aphasia, right-sided weakness, and hypoesthesia. The patient awoke with neurological symptoms on the day of admission.

Computer tomography (CT) of the head and computed tomography angiography (CTA) of the aortic arch and intracranial arteries were performed. The patient did not qualify for thrombolysis because of the occlusion of the intracranial artery and the unknown time of the onset of stroke. The diagnostics were extended to magnetic resonance (MR), and based on this, the patient qualified for immediate mechanical thrombectomy according to the DAWN trial protocol (NIHSS scale: 11; DWI MRI: 9 mL) [8].

Digital subtraction angiography (DSA) revealed the embolism material in the M1 section of the left middle cerebral artery (LMCA). The artery was recanalized via aspiration thrombectomy using the Penumbra system, and complete restoration of flow was obtained (according to the Thrombolysis in Cerebral Infarction scale, TICI scale 3) [9], as shown in Figure 1a,b.

In addition, CTA revealed the presence of carotid web changes in the left internal carotid artery (LICA), as shown in Figure 2a,b.

In the control CT scanning, an acute ischemic area in the left temporal lobe was found, as shown in Figure 3.

After the treatment, the patient demonstrated complete neurological improvement from his initial presentation of right weakness, hypoesthesia, and motor aphasia, with no residual deficit.

In additional studies, such as fasting blood sugar, LDL cholesterol, triglycerides, echocardiography, and Holter monitor, no significant pathologies were found. Duplex scan images showed a thickening of the intima–media complex in the LICA. In the secondary prevention period, double-platelet therapy (clopidogrel 75 mg and acetylsalicylic acid 75 mg) and statin treatment (atorvastatin 40 mg) were implemented.

The patient received consultations from a cardiologist and a vascular surgeon. The patient then qualified for carotid artery stenting (CAS) of the LICA, which was postponed to a later period due to the presence of an area of infarction. The angioplasty with stenting was performed 6 months later, and a carotid antiembolic “mesh” stent (Roadsaver, Terumo) was implanted into the LICA across the web with good vascular effect, as shown in Figure 4a,b.

## 3. Discussion

The literature clearly indicates that CaW is usually found in young patients under 60 years of age [10,11]. Our patient also belongs to this age group. In the case series study published by Multon et al., the median age among 11 symptomatic CaW patients was 41 years (range: 39–51 years) [12]. However, the youngest patient’s description of symptomatic CaW was reported by Kyaw et al. A 20-year-old female presented ischemic stroke symptoms related to the region of the ipsilateral middle cerebral artery [13].

Although CaW is usually associated with the female sex [1,3,14,15,16], in some studies, males and females show similarity [2,5,11].

The pathogenesis of CaW is not clearly explained. Genetic reasons, trauma mechanisms, or hormonal disorders are suspected [17]. The presentation of CaW is most often one-sided and located on the posterior wall of the internal carotid artery (ICA). In the study published by Rzepka M. et al., among 181 patients with acute ischemic stroke or TIA, CaW was found in 27, giving a prevalence of 14.9%. In 16/27, CaW was on the left side; in 9/27, on the right side; and in 2/27, bilateral [18].

Unlike typical FMD, changes in CaW cover only the intima of the artery and do not reach the media [19,20]. In this location, a small valve usually appears, where clots are formed as a result of the slowed blood flow. Such conditions may lead to embolization and ischemic stroke, especially when large intracranial vessels are involved [4,7,19,21]. This is in accordance with the results of a single-center study involving 466 patients with anterior circulation ischemic stroke. Ipsilateral CaW was more frequent in embolic stroke of unknown source than in the rest of the sample (10.7% (confidence interval 95% = 2.7–18.7) vs. 0.7% (0–1.5), *p* < 0.001). This difference remained significant after adjustment for sex, age, and vascular risk factors (odds ratio: 12.5 (2.1–72), *p* = 0.005) or after the exclusion of patients with any other bulb wall thickening (*p* = 0.025) [21].

Our patient experienced an ischemic stroke as a result of an occluded MCA. The stroke occurred on the same side as the CaW location, and no other possible causes were found.

According to Tabibian BE et al., the size and morphology of CaW may be associated with the risk of stroke. The authors concluded that such morphological features of CaW as a length ≥ 3 mm, an acute angle relative to the carotid wall, a thickness ≥ 1.85 mm, and stenosis of lumen > 50% were present more often in symptomatic patients compared with asymptomatic. However, vessel narrowing caused by CaW may be a doubtful parameter in relation to stroke occurrence [22]. Kim SJ et al. examined patients with cryptogenic stroke and found that symptomatic CaW formed less than 30% of the degree of stenosis [23].

In the performed imaging studies before the intervention, the CaW change was detected only in the CTA with morphology suitable for the features mentioned above.

In the Duplex scan images, only local non-specific thickening of the wall was found, which is consistent with other reports. Duplex scanning alone is insufficient for recognizing CaW, as it is easy to confuse this pathology with atherosclerotic plaque. Therefore, CTA remains a gold diagnostic standard and is considered the first-line diagnostic modality [16,20,24].

CaW pathology is best viewed in an oblique projection using the multiplanar reconstruction (MPR) technique. Moreover, it should be differentiated from atherosclerotic plaque, dissection, and FMD [7]. Atherosclerotic plaque is rarely limited to the posterior wall; its surface is often irregular and calcified and covers the entire circumference of the vessel. Dissection, in turn, most often includes a longer section of the artery, below and above the carotid bulb. FMD usually affects the middle section of the ICA [25,26,27].

In our case, in an acute phase of stroke, we performed an intracranial endovascular treatment. The patient was enrolled for such treatment according to the DAWN protocol, based on imaging studies and a therapeutic window between 6 and 24 h [8]. DSA showed embolism deposits in the LMCA, which were removed using aspiration thrombectomy. This technique is considered a therapeutic option for patients with stroke due to intracranial atherosclerotic and embolic large vessel occlusion. In both etiologies, the results regarding recanalization, favorable outcomes (at 90 days), and mortality (at 90 days) are comparable [28]. After the stroke episode, we commenced double-platelet therapy. This pharmacological therapy is preferred as the initial treatment in both asymptomatic and symptomatic CaW [11].

In observational studies, the risk of the recurrence of stroke in symptomatic CaW is 17% over 2 years [29]. This was also proven in a systematic review of CaW therapies published by Zhang et al. The authors identified 135 patients with symptomatic CaW, and those who were not treated with invasive techniques had a 56% rate of recurrent stroke. In the subgroup that received carotid intervention (CEA or CAS), no recurrent strokes were observed [15]. Therefore, invasive treatment should be offered to these patients. Surgical endarterectomy can be technically difficult, as there is usually no significant narrowing in the CaW location. The only advantage of this method is the histopathological evaluation of excised tissue [7].

Angioplasty with stenting is a remarkable and less invasive technique. The validity of this method was confirmed in a study of symptomatic patients with stent implantation across the CaW lesion. No patient had a recurrent stroke during the 12-month follow-up period [11,30]. In our case, a mesh stent (Roadsaver, Terumo) with a double layer dedicated to thrombogenic changes was successfully implanted [31,32].

The number of published CaW case reports is growing. Authors are beginning to pay special attention to CaW as a cause of stroke. We are currently observing significant developments of endovascular techniques and products that can be effectively used to treat both stroke and CaW lesions. Our study is a case in point of this. The extended therapeutic window after stroke allowed for aspiration thrombectomy with an excellent clinical effect. In turn, we believe that the stent treatment of CaW lesions is a much better solution than endarterectomy. Table 1 presents a summary of published CaW case reports. While some authors reported stent implantation, only two studies specified the type of stent used. Our patient was ultimately treated with the dual-layer stent, which allows appropriate radial force to be applied to arterial wall and limits the risk of thrombo-embolism complications with its dense mesh structure. Nevertheless, future registry is required to assess the safety and efficacy of mesh stent use in CaW.

## 4. Conclusions

CaW should be considered in the case of stroke resulting from unknown causes. Moreover, imaging studies performed during stroke diagnostics require precise attention to CaW changes. The case presented in this study demonstrates that CaW with subsequent stroke due to the occlusion of the MCA can be effectively treated with emerging endovascular techniques.

## Figures and Tables

**Figure 1 jcm-14-02568-f001:**
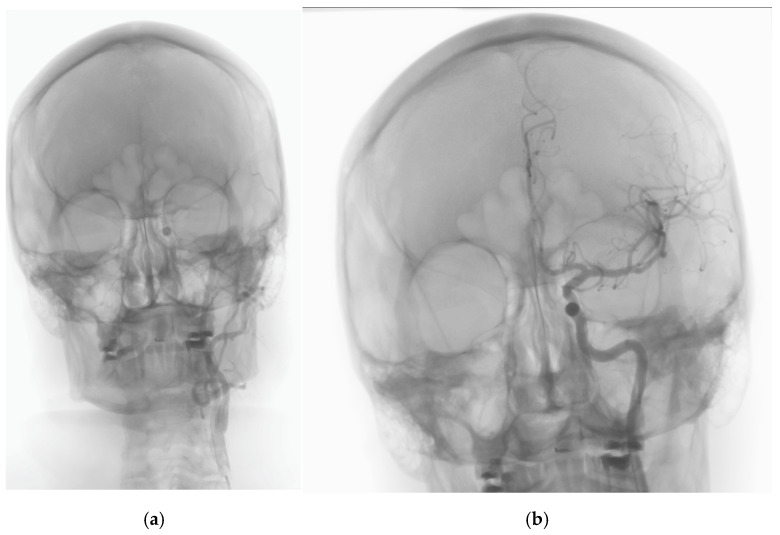
The first stage of endovascular treatment. Initial angiography of the left middle cerebral artery (LMCA) before (**a**) and after the thrombectomy (**b**), showing the achieved complete patency.

**Figure 2 jcm-14-02568-f002:**
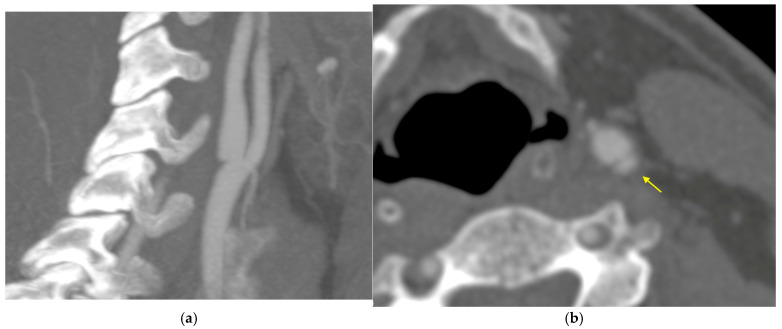
CT angiography showed a carotid web lesion (CaW) in the LICA (**a**) and the nest-like (cross-plane, indicated with an arrow) structure in the posterolateral wall of the carotid bulb (**b**), which is characteristic of CaW.

**Figure 3 jcm-14-02568-f003:**
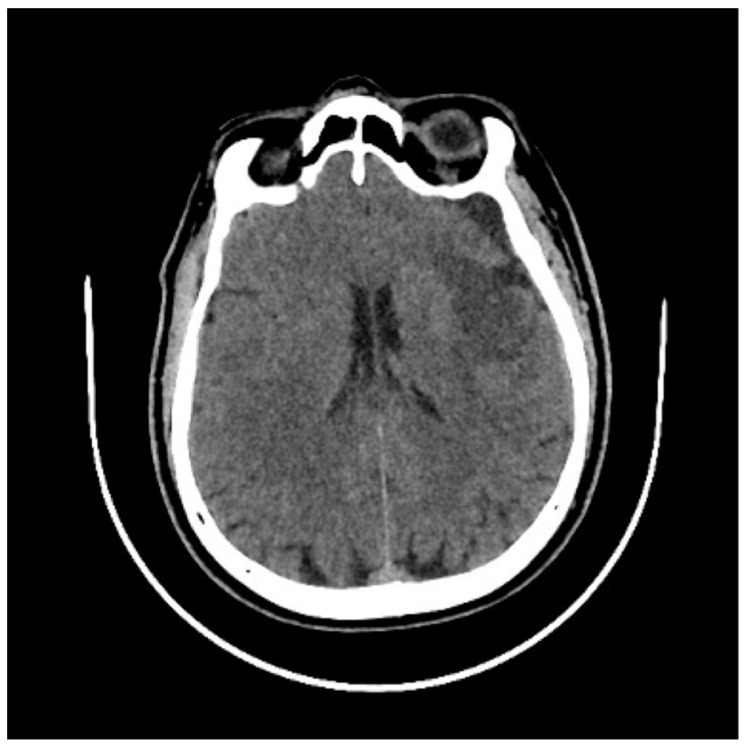
Control CT scan after 24 h of LMCA thrombectomy. An ischemic area appeared in the left temporal lobe, which required postponement of the final endovascular procedure.

**Figure 4 jcm-14-02568-f004:**
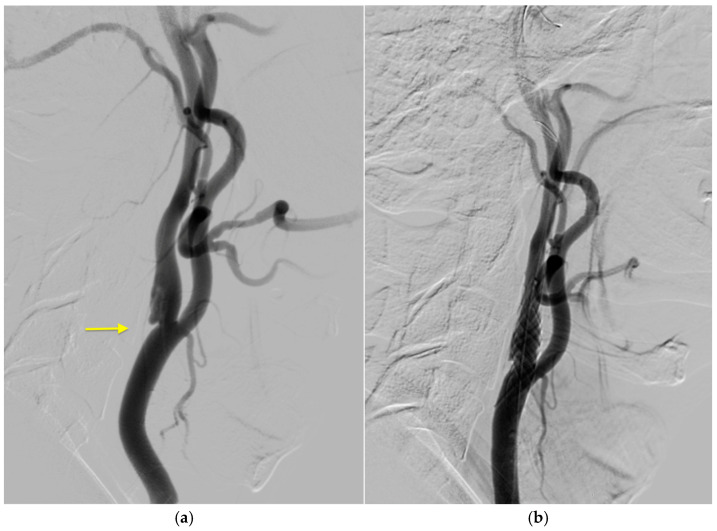
The second stage of endovascular treatment. Intraoperative angiography with confirmed CaW (indicated by the arrow) pathology (**a**) and implanted mesh stent across the lesion (**b**).

**Table 1 jcm-14-02568-t001:** Summary of published CaW case reports.

Case Reports	Age	Gender	Race	Stroke	Cardiovasc. Disease	Diabetes Mellitus	Smoking	Oral Contracept.	Medical Management of Stroke	Stroke Intervention	Carotid Intervention	Clinical
												Outcome
Yokoyama K [33]	46	M	Asian	Yes	No	No	No	-	No	tPA, MT	CEA	Residual symptoms
Takahashi T et al. [34]	50	F	Asian	Yes	No	No	No	-	No	tPA, MT Decompressive craniectomy	CEA	Partial recovery
Kyaw K et al. [13]	20	F	Caucasian	Yes	No	No	No	Yes	Yes	No	No	Poor recovery
Wojcik K et al. [35]	45	F	-	Yes, r	-	-	-	-	Yes	No	CEA	Partial recovery
	44	F	-	Yes	-	-	-	-	No	tPA, MT	CAS *	Full recovery
	52	F	-	Yes	-	-	-	-	No	tPA	No	Probable recovery
	47	M	-	TIA	-	-	-	-	Yes	No	No	Full recovery
	51	F	-	Yes	-	-	-	-	No	tPA, MT	CAS	Partial recovery
Martinez-Perez R et al. [36]	47	F	-	Yes	-	-	-	-	No	MT	CAS	Full recovery
Elmokadem AH et al. [37]	36	M	-	Yes, r	AH	No	No	-	No	MT	CAS *	Residual symptoms
	41	F	-	Yes, r	No	No	No	-	No	MT	CAS *	Residual symptoms
Our case	59	M	Caucasian	Yes	AH	No	No	-	No	MT	CAS **	Full recovery

AH—arterial hypertension, tPA—tissue plasminogen activator, MT—mechanical thrombectomy, CEA—carotid endarterectomy, CAS—carotid artery stenting, r—recurrent, * closed cell stent, ** double-layer mesh stent.

## Data Availability

All the data and models that support the findings of this study are available from the corresponding author upon reasonable request.

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
