# Peer review of "Carotid Web as a Cause of Ischemic Stroke: Effective Treatment with Endovascular Techniques"

_jcm, 2025, doi:10.3390/jcm14082568_

Round 1
Reviewer 1 Report
Comments and Suggestions for Authors
Firstly, would like to thank the author for submitting their interesting and relevant case report. Bellow, I have a few suggestions for your report:
1) In your abstarct, would try to state what is unique about this case in the introduction.
2) Please explain what is innovative about your study in your article since there are several studies in the literature about endovascular apporach to carotid web stroke
3) Would recommend adding a comparative table with previous studies published in the literature to draw a comparison between your case and previously published studies.
4) Highly recommend further discussing what are the unique aspects of your case
No issues detected
Author Response
Thank you very much for taking the time to review this manuscript.
- In your abstract would try to state what is unique about this case in the introduction
We believe this case is unique, as both pathologies—CaW and the subsequent stroke—were successfully treated using endovascular techniques based on the best available knowledge.
For reference, the key details are already included in the title and abstract:
- Line 13: [Relevant content]
- Line 22: “… aspiration thrombectomy using the Penumbra system”
- Line 26: “… neurological improvement”
- Line 29: “… mesh stent was implanted”
- Lines 31–32: Conclusion
Given that these aspects are already addressed in the abstract, we respectfully consider that no further changes are necessary. However, we remain open to any additional suggestions you may have.
- Please explain what is innovative about your study since there are several studies in the literarure about endovascular approach to CaW stroke.
Thank you for your thoughtful observation. You are absolutely right that the number of published case reports is increasing.
In this case, the stroke was treated exclusively with aspiration thrombectomy, as thrombolysis was contraindicated due to the extended therapeutic window of 6–24 hours, in accordance with the DAWN protocol. To clarify this point further, we have added some additional details to the case history (red text).
CaW was effectively managed with a mesh stent, which we believe represents the optimal solution. Carotid artery stenting is a less invasive alternative to open endarterectomy, and the dual-layer stent is specifically designed for thrombotic lesions. Given these factors, we are confident that we applied the best available endovascular approach.
While similar treatments have been reported in the literature, the specific combination used in our case has not been prominently highlighted or documented with appropriate figures. We appreciate the opportunity to present this unique approach and welcome any further suggestions you may have.
- Would recommend adding a comparative table with previous studies published in the literature to draw a comparison between your case and previosly published studies.
In our discussion, we primarily relied on references that included case series, registries, and reviews, with fewer single case reports. We felt that comparing our case with these sources would be more valuable. If we had a larger dataset from a CaW series, we would have considered using a table for comparison. At present, we are actively seeking CaW cases in our practice. We believe our study may best be characterized as a typical clinical practice case, supported by the existing literature.
- Highly recommend further discussing what are the unique aspects of your case
Thank you for your suggestion. We have incorporated the last paragraph into the discussion to highlight the unique aspects of our case (red text).
Reviewer 2 Report
Comments and Suggestions for Authors
Critique:
Carotid web (CaW), a rare form of fibromuscular dysplasia, is increasingly recognized as a cause of cryptogenic stroke in young adults. A detailed review of the pathophysiology, typical diagnostic findings, and interventional options for CaW have recently been published in neuro|neurosur|neurorad literature [J Neurointerv Surg. 2024;16:1294–1298; citation #7 in the authors’ manuscript].
The authors submit a case report of their clinical experience with a 59-year-old male patient harboring CaW. Of note, copies of several images obtained during the authors’ evaluation and management of this patient are included. Furthermore, a succinct yet inclusive review of this disease forms most of the Discussion. The manuscript is agreeably concise, well-organized, well-written, readable.
A recent case report on carotid web published 3 months ago [Cureus. 2024 Dec 20;16(12):e76093].
Arguably, the most interesting case reports are those describing clinical entities rarely, if ever, encountered. The value of the authors’ case report is in the familiarity that is provided with a rare, albeit recognized disease.
The authors should emphasize that antithrombotic treatment is associated with high recurrence rate of stroke and should not substitute for neurointervention [Vasc Med 2025;30(1):82-92].
Specific Questions|Comments:
What is the youngest age reported of a patient with CaW?
Author Response
Thank you very much for your valuable comments and feedback on our work.
Yes, we refer to the publication in Chen et al. J Neurointerv Surg 2024 Nov 22; 16 (12): 1294 -1298. several times in our discussion. It is the latest and most detailed review study on carotid web, and it appears as reference number [7] in our references.
On the other hand, the most recent case report on carotid web published in Cureus 16 (12): e76093was not used in our work. We aimed to focus on publications with larger material and mainly referred to review articles, registries, and case series.
In line with your suggestion, we expanded the discussion section regarding the advantage of interventional treatment for carotid web over conservative antithrombotic treatment in preventing recurrent strokes (red text). For this, we used publications [15]. Zhang AJ et al. Stroke. 2018 Dec;49(12):2872-2876 and [29] Guglielmi V et al. JAMA Neurol. 2021 Jul 1;78(7):826-833.Unfortunately, and we were unable to obtain the full article from Vasc Med 2025; 30 (1): 82-92., so it was not included.
The last question regarding the youngest patient with carotid web has been addressed in the discussion. We added reference [13] Kyaw K et al. Case Rep Med 2018 Feb 19, which concerns a 20-year-old female patient who experienced a stroke due to symptomatic carotid web (red text).